# The Interplay between Structural Inequality, Allostatic Load, Inflammation, and Cancer in Black Americans: A Narrative Review

**DOI:** 10.3390/cancers16173023

**Published:** 2024-08-30

**Authors:** Ashanda R. Esdaille, Nelson Kevin Kuete, Vivian Ifunanya Anyaeche, Ecem Kalemoglu, Omer Kucuk

**Affiliations:** 1Department of Urology, Emory University School of Medicine, Atlanta, GA 30322, USA; 2Division of Urology, Atlanta Veteran’s Affairs Medical Center, Decatur, GA 30033, USA; 3Department of Hematology and Medical Oncology, Winship Cancer Institute, Emory University School of Medicine, Atlanta, GA 30322, USA

**Keywords:** healthcare, disparities, stress, inflammation, cancer

## Abstract

**Simple Summary:**

Racial healthcare disparities, driven by adverse living conditions, environmental factors, and systemic biases, significantly affect Black Americans, and they are associated with increased oxidative stress, inflammation, and, in turn, chronic diseases like cardiovascular disease and cancer. Addressing these disparities requires comprehensive systemic reforms alongside additional strategies such as increased exercise, stress reduction, and anti-inflammatory diets. This review highlights the relevant literature and aims to encourage further research on this important topic.

**Abstract:**

The impact of racial healthcare disparities has been well documented. Adverse social determinants of health, such as poverty, inadequate housing, and limited access to healthcare, are intricately linked to these disparities and negative health outcomes, highlighting the profound impact that social and economic factors have on individuals’ overall well-being. Recent evidence underscores the role of residential location on individual health outcomes. Despite the importance of a healthy lifestyle, regular physical activity, balanced nutrition, and stress management for favorable health outcomes, individuals living in socioeconomically disadvantaged areas may face obstacles in achieving these practices. Adverse living conditions, environmental factors, and systemic biases against Black Americans perpetuate allostatic load. This, compounded by decreased physical activity and limited healthy food options, may contribute to increased oxidative stress and inflammation, fundamental drivers of morbidities such as cardiovascular disease and cancer. Herein, we perform a narrative review of associations between healthcare disparities, chronic stress, allostatic load, inflammation, and cancer in Black Americans, and we discuss potential mechanisms and solutions. Additional research is warranted in the very important area of cancer disparities.

## 1. Introduction

Although there has been a decline in cancer-related deaths among Black individuals in the United States over the past two decades, Black men and women still have elevated cancer mortality rates compared to other U.S. demographic groups [1]. The proposed reasons for this cancer disparity are multifactorial, but they include systems issues such as dissimilar screening, staging, and management, as well as patient-level factors such as inadequate access to care, acquired epigenetic alterations, and modifiable environmental risk factors or exposures [2].

Previous research has demonstrated that one’s zip code exerts a paramount influence on an individual’s health outcomes [3,4,5,6]. Many Black American communities have been subject to racist, historical policies such as “redlining” resulting in the funneling of these populations into specific neighborhoods or zip codes that are characterized by increased exposure to environmental pollution, lack of green spaces, increased prevalence of food swamps and deserts, and inadequate healthcare access [7,8,9]. Further, these neighborhoods are often disproportionately affected by chronic psychological and physiological stressors such as gun violence and high rates of drug, tobacco, and alcohol abuse [10]. Neighborhood redlining, and the subsequent disadvantages, has been associated with late-stage diagnosis for breast and colorectal cancer with significant implications on cancer-specific mortality [11,12,13]. A population-based cohort study found that compared with living in the most affluent neighborhoods, living in disadvantaged neighborhoods was associated with a 1.5× higher risk of cancer-related deaths across breast, lung, colorectal, and prostate cancers [14]. However, the association between structural inequality, socioeconomic disadvantages, and cancer biology is understudied. Here, we perform a narrative review of the relevant literature and discuss the potential mechanisms of, and potential solutions to, this issue. We reviewed the English literature in PubMed from 1990 to 2024 and included papers addressing ethnic disparities in healthcare and cancer. We believe this review will generate significant interest in the scientific community and may provide perspectives on the interplay between socioeconomic disadvantages, the environment, and biological factors. In the future, our group plans to conduct additional studies to elucidate the associations between health inequity and cancer molecular mechanisms.

## 2. Biological Mechanisms Linking Inflammation and Carcinogenesis

There has been a longstanding recognition of a functional connection between chronic inflammation and cancer, where inflammation has been implicated as the driver of carcinogenesis [15,16] in many solid tumors. Recent evidence indicates that such inflammation influences nearly every stage of cancer, including development, metastasis, drug resistance, and recurrence, by causing genomic instability, fostering the self-renewal of cancer stem-like cells, and promoting angiogenesis. Currently, tumor-associated inflammation is acknowledged as the seventh biological feature of cancer. The process of inflammation can lead to the production of reactive oxygen species (ROS), which can damage DNA and promote the growth and survival of cancer cells [17]. In addition, it can contribute to the accumulation of genetic mutations and epigenetic changes that promote cancer development [18]. This process leads to the activation of pro-inflammatory pathways, including the production of cytokines such as interleukin-6 (IL-6) and tumor necrosis factor-alpha (TNF-alpha), which can promote tumor growth and metastasis. Within the tumor microenvironment, there is balance among the various cellular components to maintain homeostasis. In the setting of malignant transformation, there is disequilibrium where the immune milieu can have both pro-cancerous and anti-cancerous effects [19], contributing to the evolution of a more aggressive tumor [20,21,22,23,24].

While inflammation can be pro-carcinogenic, the data suggest that the predominance of certain immune cell types within a primary tumor microenvironment can be a good prognostic indicator. For instance, a density of tumor-infiltrating T-cells (TILs) is associated with improved patient cancer-specific survival [25]; conversely, an abundance of regulatory T-cells is linked to poor prognosis [26,27]. M1 macrophages secrete interferon-gamma (IFN-γ), which has an anti-cancerous effect, whereas tumor-infiltrating M2 macrophages are pro-cancerous and are associated with tumor growth and metastasis [28]. These inflammatory mediators directly affect both cancer and stromal cells and contribute to several hallmarks of cancer, such as the promotion of the epithelial-to-mesenchymal transition (EMT) and metastasis of cancer cells [29]. 

For example, a similar process is observed in prostate cancer, where in the setting of chronic inflammation, tumor-associated macrophages secrete multiple cytokines such as tumor necrosis factor (TNF), IL-7, IL-2, and inflammatory proteins, resulting in the activation of NF-kB signaling and TGF-β (inducing EMT) which alter the cancer microenvironment. This cascade subsequently activates growth factors including fibroblast growth factor (FGF), transforming growth factor-β (TGF-β), and vascular endothelial growth factor [30]. The release of inflammatory mediators into the extracellular matrix activates stromal cells which contribute to a cytokine-rich and inflamed microenvironment that nurtures tumor cells towards prostate cancer development and metastasis [31]. The factors that influence inflammation and, in turn, potentiate a pro-carcinogenic tumor microenvironment are vast, including viruses or infections, inflammatory diseases, and environmental factors, such as obesity, high-fat diets, and stress [32,33,34,35].

## 3. Stress, Discrimination, Inflammation, and Disease

Allostatic load (AL) refers to the cumulative “wear and tear” on the body due to chronic stress and is an indicator of accelerated physiological aging. This concept was developed by McEwen and Stellar in 1993, proposing stress as a contributor of disease initiation and progression [36]. These stressors can be physical, psychological, or environmental in nature, leading to the activation of both the hypothalamic–pituitary–adrenal (HPA) axis and the sympathetic nervous system with the release of cortisol and epinephrine [37]. While acute stress has favorable implications as it relates to wound healing, chronic stress can lead to the dysregulation of these physiological systems, manifesting in alterations in hormone levels, inflammation, immune system dysfunction, metabolic changes, and cardiovascular changes. Over time, the accumulated effects of these physiological changes can contribute to an increased risk of various comorbid conditions.

Allostatic load is quantified by combining biomarkers, such as blood pressure, cholesterol, and C-reactive protein levels, from various physiological systems, such as the metabolic, immune, cardiac, and hematologic systems. The construct validity of AL has been well established, with evidence indicating that allostatic overload correlates with an increased risk of cardiovascular disease, mental illnesses, and cancer [38,39,40,41,42,43]. In a recent cohort study, the AL score incorporated eleven factors: three cardiovascular (systolic blood pressure, diastolic blood pressure, and pulse rate), one inflammatory (CRP), six metabolic (high-density lipoprotein, waist-to-hip ratio, abnormal cholesterol, triglyceride, HbA1c, creatinine), and one medication factor. Each factor was assigned a binary value (1 or 0) based on clinical risk thresholds and the total score, ranging from 0 to 11 units, indicated the level of AL. Increased scores reflected increased levels. A higher AL score was associated with a significantly increased risk of breast cancer, where the risk increased by 5% per one AL unit increase [43]. Similar associations were identified in patients with advanced non-small-cell lung cancer where higher AL scores were associated with adverse social determinants of health and worsened overall mortality [44]. 

Black Americans, in comparison to their White American counterparts, are disproportionately impacted by these effects of structural inequality and racial discrimination, leading to increased AL scores [45,46,47]. As of 2021, Black Americans are the second largest minority population in the United States, following the Hispanic/Latino population [48]. Despite their population size, Black Americans are minoritized and often part of a marginal subculture due to neighborhood-level and individual-level socioeconomic barriers. These barriers manifest across various sectors including environmental engagement, national park visitation, cultural expression, educational and community navigation, and health disparities [49,50,51].

Allostatic load reflects the marginalization of minoritized Americans with a significant impact on all-cause and cancer-specific mortality. Stress or depressive symptoms have been identified as potential mediators for the effects of perceived discrimination and cancer. In a sample of 1363 Black American adults, higher levels of discrimination were associated with increased stress and depressive symptoms, subsequently elevating the likelihood of smoking and other behavioral risk factors that have been linked to cancer [52].

In addition to the psychological or behavioral consequences of systemic racism and structural inequality, discrimination is associated with inflammation and has a subsequent impact on disease. Several studies have observed positive associations between racial discrimination and elevated inflammatory biomarkers such as C-reactive protein (CRP), IL-6, and E-selectin levels [53,54,55,56,57,58]. In Van Dyke et al.’s study, linear regression analyses were used to examine the associations among SES, discrimination, race, education, and CRP after controlling for age, gender, racial and gender discrimination, financial and general stress, body mass index, smoking, sleep quality, and depressive symptoms. After adjusting for sociodemographic variables and racial and gender discrimination, a notable interaction was found between race, education, and SES discrimination (*p* = 0.019). In models adjusted and stratified by race and education level, SES discrimination showed a positive association with increased CRP levels in Black Americans with higher education (β = 0.29, *p* = 0.018). However, this association was not observed in Black Americans with lower education (β = −0.13, *p* = 0.32), nor in lower-educated (β = −0.02, *p* = 0.92) or higher-educated Whites (β = −0.01, *p* = 0.90) [55]. These data may highlight the multifaceted and intersectional experiences of Black Americans. Individuals with a higher socioeconomic status or educational level may experience various forms of discrimination such as wealth gaps, occupational segregation, or implicit biases.

Elevated CRP levels and metabolic dysregulation, as a result of high levels of stress attributed to everyday and major life discrimination, have been found to be major contributors to racial health disparities [59]. Cuevas et al. conducted a systematic review of sixteen cross-sectional and nine longitudinal studies focusing on unfair treatment or discrimination and its correlation with systemic inflammation [60]. Studies identified robust associations between discriminatory experiences, whether acute or chronic, and immune system dysfunction, marked by elevated inflammation levels. These findings align with the conclusion that heightened inflammation, identified as a crucial risk factor for various diseases, is a consistent outcome of discrimination. Chronic stress profoundly impacts susceptibility to cancer by suppressing protective immunity. For example, an anxious behavioral phenotype is associated with greater chronic stress and immune suppression, which correlates with a higher tumor burden [61,62,63]. This suppression involves a reduced expression of key immune molecules, hindering the recruitment and function of protective T-cells. Moreover, chronic stress increases regulatory/suppressor T-cell numbers and intensifies immune suppression overall, contributing to the accelerated emergence and progression of pre-cancerous and cancerous lesions.

In a study by Boyle et al., neighborhood disadvantage was associated with a higher expression of stress-related genes, which could contribute to an increased risk of aggressive prostate cancer. Two neighborhood deprivation metrics (Area Deprivation Index [ADI] and validated Bayesian Neighborhood Deprivation Index) as well as the Racial Isolation Index (RI) and historical redlining were applied to participants’ addresses. A total of 105 stress-related genes were evaluated with each neighborhood metric using linear regression, adjusting for race, age, and year of surgery. Genes in the Conserved Transcriptional Response to Adversity (CTRA) and stress-related signaling genes were included. Notably, the expression of several stress-related genes in prostate tumors was higher among men residing in disadvantaged neighborhoods [64]. These data reinforce the imperative to recognize and address the impact of discrimination, inequality, and resulting stress on tumor profiles as well as on innate and adaptive immunity, emphasizing the broader implications for overall health, particularly cancer, outcomes (Table 1 and Figure 1).

## 4. Strategies to Reduce Inflammation and Improve Health Outcomes

The exposure to chronic stressors suffered by those residing in underserved neighborhoods leads to disparities in life expectancy, morbidity, and the rates of chronic diseases, further emphasizing the impact of allostatic load on overall health and the potential effectiveness of neighborhood-directed interventions. Strategies that reduce chronic inflammation, such as healthy diets, regular exercise, and stress reduction techniques, could help to reduce the risk of cancer and other chronic diseases.

### 4.1. Stress Reduction Interventions

Stress management plays a crucial role in mitigating the negative effects of stress on cancer, as evidenced by numerous studies exploring stress reduction techniques and their impact on cancer outcomes. These interventions investigated in cancer patients strive to regulate psychological tension and physiological arousal through methods such as muscle relaxation training, deep breathing, yoga, Tai Chi, massage, acupuncture, and biofield therapies. Alternatively, some interventions focus on raising awareness and fostering a non-judgmental mindset towards stress-inducing thoughts, employing mindfulness-based stress reduction techniques [61,65]. A mindfulness-based stress reduction program, a program that includes meditation, yoga postures, and relaxation, demonstrated a decrease in stress levels among breast cancer and prostate cancer patients. Various studies exploring yoga interventions also exhibited notable stress reduction and in some cases positive effects on symptoms linked to stress, such as fatigue, sleep disturbances, hot flashes, and the overall quality of life. 

Furthermore, these stress reduction interventions might be beneficial during the cancer treatment and survivorship follow-up processes [61,66,67]. Tai Chi has been shown to provide benefits to cancer survivors related to physical deconditioning, cardiovascular disease risk, and psychological stress [68,69]. In a randomized control trial by Mustian et al., women that completed breast cancer treatment and participated in Tai Chi exhibited noteworthy enhancements in functional capacity, aerobic capacity, muscular strength, flexibility, self-esteem, bone health, immune function, and quality of life [70]. 

While certain pharmacological interventions, such as the use of statins and angiotensin-converting enzyme inhibitors, demonstrate a reduction in inflammation, indicated by decreased CRP concentrations, none of the pharmaceutical agents with anti-inflammatory effects are currently approved for the continuous treatment of persistent inflammation. In contrast, lifestyle behavioral interventions such as exercise, stress management, smoking cessation, and modifications in food/dietary intake may yield clinically significant advantages in improving inflammation over time [71,72]. 

### 4.2. Improved Physical Activity 

Several large studies on population-based cohorts have investigated the effects of exercise on chronic inflammation, including the British Regional Heart Study [73], the Third National Health and Nutrition Examination Survey (NHANES III) [74,75], the Cardiovascular Health Study [76], the Men’s Health Professionals Follow-up Study, the Nurses’ Health Study II [77], the MacArthur Studies of Aging [78], the Multi-Ethnic Study of Atherosclerosis (MESA) [79], the CHIANTI study [80], and the Health, Aging, and Body Composition Study (Health ABC) [81]. These studies provide data suggesting an inverse association between CRP concentration and physical activity [71]. In NHANES III, physical activity more than 22 times per month was linked to a 37% decrease in the risk of elevated CRP compared to engaging in activity less than three times per month. Even sporadic physical activity in the British Regional Heart Study was associated with a 39% decrease in CRP in middle-aged men. Therefore, physical activity is recognized as an important strategy for reducing chronic inflammation and, in turn, the risk of chronic diseases such as cancer [71,73,75]. There are also data on the correlation between other inflammatory markers and physical activity. In the Health ABC study, there was a trend towards lower IL-6, TNFα, and CRP levels with increased physical activity. Among older men, both IL-6 and CRP concentrations showed negative associations with the reported hours per year of moderate and strenuous physical activity. In addition, the lowest concentrations of both CRP and IL-6 were found in individuals engaged in high levels of recreational activity. Relatively higher levels of light-intensity activity and walking were associated with decreased levels of soluble tumor necrosis factor receptor-2 (sTNFR2). In the Health Professionals Follow-up and Nurses’ Health Studies, a significant dose–response relationship was observed between physical activity and inflammatory markers. Individuals running more than four hours per week had 4% lower soluble tumor necrosis factor receptor 1 (sTNFR1) and sTNFR2, 6% lower IL-6, and 49% lower CRP compared to those running less than half an hour per week. Additionally, in the ATTICA study, physically active individuals with metabolic syndrome demonstrated 30% lower IL-6, 15% lower TNFα, 19% lower serum amyloid-A (SAA), and 15% lower white blood cell (WBC) counts compared to sedentary individuals [71,77,78,81,82,83]. These results strongly suggest that physical activity is associated with lower systemic inflammation, potentially reducing the risk of chronic disease and cancer.

### 4.3. Adherence to Anti-Inflammatory Diets 

A balanced diet, rich in diverse healthy dietary components, is also an important strategy to control chronic inflammation, as it directly contributes to immune homeostasis. Diet indirectly influences immune balance through interactions with gut microbiota and their metabolites, and unhealthy diet has the potential to shift the immune balance towards pro-inflammation, leading to systemic inflammatory responses. Animal trials have demonstrated that foods rich in saturated fats and sugar can trigger inflammatory conditions through microbial processes, including the stimulation of T-helper 17 (TH17) cells. Conversely, tryptophan metabolites exert anti-inflammatory effects by activating the aryl hydrocarbon receptor in lymphoid tissues and promoting the generation of regulatory T (Treg) cells. A high intake of dietary fiber encourages the growth of microbial populations that produce short-chain fatty acids. These fatty acids activate Treg cells, which may lead to the suppression of inflammation [84,85,86,87,88,89,90,91].

Diets associated with chronic diseases are characterized by elevated levels of processed foods, red meat, high-fat dairy products, high-sugar foods, pre-packaged foods, and low dietary fiber. This type of diet induces pro-inflammatory changes, such as the activation of pattern recognition receptors (PRRs) such as toll-like receptor (TLR) 4, subsequently initiating the release of pro-inflammatory mediators cyclooxygenase 2 (COX2), tumor necrosis factor-α (TNF-α), interleukin-1β (IL-1β), interleukin-6 (IL-6), interleukin-8 (IL-8), interleukin-12 (IL-12), and interferon-γ (IFN-γ). Such dietary-induced inflammation associated with impaired immune functions and gut dysbiosis may consequently lead to inflammatory disorders [91,92,93]. On the other hand, the Mediterranean diet is centered around the intake of substantial quantities of vegetables, fruits, cereals, legumes, nuts, and fish, with olive oil serving as the primary culinary fat. There is a negative association with the serum markers of inflammation and adherence to a Mediterranean diet, where unsaturated fatty acids, integral components of Mediterranean diets, correlate with an anti-inflammatory phenotype [86,94,95,96].

## 5. Impact of Structural Inequality on Cancer Prevention Strategies

Although the evidence underscores the importance of physical activity, stress management, and healthy, anti-inflammatory diets in reducing chronic diseases and malignancies, many underserved, Black Americans encounter barriers to achieving these goals. Among individuals aged ≥ 20 years in 2017–2018, the prevalence rates for obesity were the highest in Black American people (49.6%) compared with non-Hispanic White Americans, (42.2%) [97]. In 2018, Black Americans were 20% less likely to participate in physical activity than their White counterparts [98]. While barriers to physical activity are vast including intra- and interpersonal barriers, numerous qualitative and quantitative studies have identified community and environmental barriers to physical activity such as safety concerns including physical harm [99], gun violence or gang-related activity [100], or living in neighborhoods perceived to be unsafe to walk in after dark [101,102]. Further, additional studies have identified positive associations with a lack of sidewalks or neighborhood facilities and decreased physical activity [103,104,105]. Along with decreased physical activity, poor-quality diets and inadequate access to healthy food also contribute to the prevalence of obesity in the Black American community. Neighborhoods with a high proportion of Black Americans are often characterized by an increased density of fast-food restaurants [106] and less access to supermarkets in comparison to predominantly White neighborhoods [107,108]. 

Individuals with lower socioeconomic and educational backgrounds and racial and ethnic minorities are significantly impacted by low health literacy. According to the 2003 National Assessment of Adult Literacy, it was found that 58% of Black Americans possess only basic or below basic health literacy levels, as opposed to 28% of non-Hispanic Whites. Many studies have identified low health literacy as a crucial factor contributing to racial and ethnic disparities in health behaviors like smoking and adherence to HIV treatments, the accessibility of healthcare resources, and overall health outcomes [109,110].

Factors like limited educational opportunities, systemic racism, mistrust in the health system, and a lack of culturally appropriate health information are exacerbating factors. Additionally, education, crucial for enhancing health literacy, is unevenly distributed across racial lines due to factors like racial residential segregation and discrimination, limiting quality educational access for Black Americans [110]. Furthermore, historical abuses, such as the Tuskegee Syphilis Study, have fostered a deep mistrust among Black Americans towards the healthcare system, impacting their health literacy and engagement with healthcare services. This mistrust, coupled with healthcare providers often underestimating the health literacy levels of Black American patients, hinders effective communication and tailored health interventions [110,111].

## 6. Conclusions and Future Directions

Chronic stress fosters inflammation—an established driver of carcinogenesis. Strategies such as exercise, mindfulness-based stress reduction, and the adoption of anti-inflammatory diets emerge as effective tools to mitigate the adverse effects of chronic inflammation and reduce the risk of chronic diseases, including cancer. However, many Black Americans may experience barriers to achieving healthy lifestyles, emphasizing the need for neighborhood-specific interventions to directly affect disparities on the neighborhood level. Further, there is an urgent need for systemic policy changes to address systemic racism and its deleterious effects. Accurately measuring and defining disparities that can be attributed to racist policies can be the first step in creating policy change to ensure equitable access to healthcare and to create environments conducive to a healthy lifestyle. In the future, our group plans to conduct additional studies to investigate the associations between structural inequality, prostate tumor-related inflammation, and immune-based mechanisms of disease. Further exploring these connections could uncover critical insights that could drive more equitable and effective prostate cancer prevention and treatment. 

There are limitations of this review because it was not a systematic review and may have missed some important information. Furthermore, not all papers measured the association between structural inequality, cancer-related AL, inflammation, stress, and the lack of cancer prevention strategies. It is also unknown whether structural inequality is always stronger in Blacks than in Whites. There may be structural inequality in other racial and ethnic groups as well with similar issues of high stress, high inflammation, and risk of cancer. Future research should investigate racial and ethnic disparities with respect to different types of cancers in various populations.

## Figures and Tables

**Figure 1 cancers-16-03023-f001:**
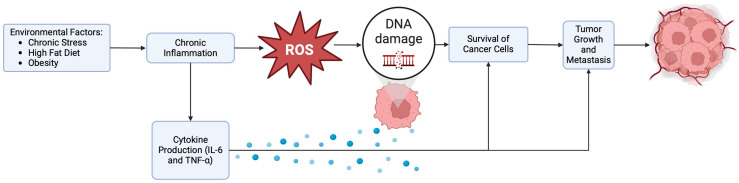
Environmental factors, inflammation, oxidative stress, and cancer.

**Table 1 cancers-16-03023-t001:** Summary of selected studies on discrimination, stress, inflammation, and cancer.

Authors	Study	*n*	Methods	Results
Cuevas AG et al. (2014) [52]	Discrimination, Affect, and Cancer Risk Factors among African Americans	1363	Nonparametric bootstrapping procedures, adjusted for sociodemographics, were used to assess mediation.	Discrimination may impact certain behavioral cancer risk factors by increasing levels of stress and depressive symptoms.
Doyle DM and Molix L (2014) [54]	Perceived discrimination as a stressor for close relationships: identifying psychological and physiological pathways	592	Secondary data from the Midlife in the United States II (MIDUS II): Milwaukee African American Sample were analyzed.	Discrimination was indirectly associated with increased emotion dysregulation through stressor appraisals and directly associated with increased inflammation (IL-6, e-selectin, and CRP).
Van Dyke ME et al. (2017) [55]	Socioeconomic status discrimination and C-reactive protein in African-American and White adults	401	Population-based cohort in the Southeastern United States. SES discrimination was self-reported with a modified Experiences of Discrimination Scale, and CRP levels were assayed from blood samples.	SES discrimination is an important discriminatory stressor, and it is associated with elevated CRP levels specifically among higher-educated African Americans.
Lewis, TT et al. (2010) [56]	Self-reported Experiences of Everyday Discrimination are associated with Elevated C-Reactive Protein levels in older African-American Adults	296	African American adults from the Minority Aging Research Study (MARS) were included if they had completed the baseline MARS evaluation and had serum available for the measurement of CRP. They were assessed with everyday discrimination by the 9-item Everyday Discrimination Scale.	Self-reported experiences of everyday discrimination are associated with higher levels of CRP in older African American adults.
Ong AD et al. (2017) [57]	Everyday unfair treatment and multisystem biological dysregulation in African American adults	233	Perceptions of everyday unfair treatment were measured by a questionnaire. The allostatic load index was computed as the sum of 7 separate physiological system risk indices.	Everyday mistreatment was associated with higher allostatic load.
Stepanikova I et al. (2017) [58]	Systemic Inflammation in Midlife: Race, Socioeconomic Status, and Perceived Discrimination	1054	Data were obtained from the Survey of Midlife in the U.S. The main outcome measures were fasting blood concentrations of C-reactive protein, interleukin 6, fibrinogen, and E-selectin. For each biomarker, series of multivariate linear regression models were estimated for the pooled sample and separately for Blacks and Whites.	Race, SES, and perceived discrimination contribute to inflammation. Also, this study suggested that inflammation-reducing interventions should focus on Blacks and individuals facing socioeconomic disadvantages, especially low education.
Boen C (2020) [59]	Death by a Thousand Cuts: Stress Exposure and Black–White Disparities in Physiological Functioning in Late Life	7280	The data from the Health and Retirement Study (HRS) (2004–2012) were used. Stepwise ordinary least squares (OLS) regression models were applied to examine the prospective associations between multiple stressors and CRP and metabolic dysregulation.	Blacks experienced more stress than Whites, and stress exposure was strongly associated with CRP and metabolic dysregulation.
Cuevas, AG et al. (2020) [60]	Discrimination and systemic inflammation: A critical review and synthesis	28 studies recruited a total of 60,039 respondents	Preferred Reporting Items for Systematic Reviews and Meta-analysis protocol for scoping reviews (PRISMA-ScR) were followed.	The research reviewed suggested that experiences of discrimination, both acute and chronic, can dysregulate immune function, characterized by elevated levels of inflammation.
Shen, J et al. (2022) [35]	Association of Allostatic Load and all Cancer Risk in the SWAN Cohort	3015 women	Acquired the data from the Study of Women’s Health Across the Nation (SWAN), a multi-center study of women’s health through menopausal transition comprising a baseline evaluation and ten waves of following annual evaluations.	Individual biomarkers of the AL score and higher levels of triglyceride and CRP were associated with an increased risk of cancer.
Guan, Y et al. (2023) [43]	Association between Allostatic Load and Breast Cancer Risk: Cohort Study	5701 women	The study population was identified from the UK Biobank, a prospective cohort study containing in-depth genetic and health information. The AL score included a total of eleven factors, including three cardiovascular (SBP, DBP, PR), one inflammatory (CRP), six metabolic (HDL, waist-to-hip ratio, abnormal cholesterol, TG, HbA1c, creatinine), and one medication factor.	Compared with women in the low-AL group, those in the high-AL group had a 1.17-fold increased risk of breast cancer.
Boyle, J et al. (2024) [64]	Neighborhood Disadvantage and Prostate Tumor RNA Expression of Stress-Related Genes.	268	This cross-sectional study leveraged prostate tumor transcriptomic data for African American and White men with prostate cancer who received radical prostatectomy at the University of Maryland Medical Center. Using addresses at diagnosis, 2 neighborhood deprivation metrics (Area Deprivation Index [ADI] and validated Bayesian Neighborhood Deprivation Index) as well as the Racial Isolation Index (RI) and historical redlining were applied to participants’ addresses. A total of 105 stress-related genes were evaluated with each neighborhood metric using linear regression, adjusting for race, age, and year of surgery. Genes in the Conserved Transcriptional Response to Adversity (CTRA) and stress-related signaling genes were included.	African American participants experienced greater neighborhood disadvantage than White participants. In this cross-sectional study, the expression of several stress-related genes in prostate tumors was higher among men residing in disadvantaged neighborhoods. This study is one of the first to suggest associations of neighborhood disadvantage with prostate tumor RNA expression.

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
