# Peer review of "The Interplay between Structural Inequality, Allostatic Load, Inflammation, and Cancer in Black Americans: A Narrative Review"

_cancers, 2024, doi:10.3390/cancers16173023_

Round 1

Reviewer 1 Report (Previous Reviewer 1)

Comments and Suggestions for Authors

The authors have responded to my comments well and have done very thorough edits that have improved the manuscript. I have no additional comments.

Author Response

REVIEWER 1

No further changes are requested.

RESPONSE: We thank the reviewer.

REVIEWER 2

Great work. However authors still need to better provide their efforts and quality of their work, instead of simply summarizing the multiple comments as very simple sentences. Also please do not highlight the revised part, please use Track change.  Also in the responses to comments, please provide page, and line of the revised part. so that we can easily match up the revised part corresponding to the reviewer comments. 

RESPONSE: We have used tracking and we provide page and line number.

I suggest the revision below:

  1. Barriers to Improving Health Outcomes – change title to “Impact of Structural Inequality on Cancer Prevention Strategies”

RESPONSE: We have changed the title to Section 5 to “Impact of Structural Inequality on Cancer Prevention Strategies” as suggested by the reviewer. (Page 9, line 292)

Limitation. please clearly address the limitation of the current review

1) as this is not the systematic review, may missing some information from the systematic review

2) Previous selected articles used in this study did not really measure the association between structural inequality and cancer-related AL, inflammation, stress, and lack of cancer prevention strategies; furthermore, it is unknown these structural inequality is always stronger in Black than Whites. There may be structural inequality in other races as well with the same issues of high stress, high inflammation, and risk of cancer. So please address that future studies also need to consider to examine structural inequality and cancer related health outcomes in other races as well. 

RESPONSE:

We have added a paragraph at the end of the last section addressing the limitations of our review and future studies to address the issues as suggested by the reviewer. (Page 10, lines 345-352)

We thank the reviewers for allowing us to improve our manuscript.

Reviewer 2 Report (Previous Reviewer 2)

Comments and Suggestions for Authors

Great work. Howewer authros still need to better provide their efforts and quality of their work, instead of simply summarizing the multiple comments as very simple sentences. Also please do not highlight the revised part, please use Track change,  Also in the responses to comments, please provide page, and line of the revised part. so that we can easily match up the revised part corresponding to the reviewer comments. Thank you. 

I suggest the revision below:

5. Barriers to Improving Health Outcomes

please check this subtitle as 

Impact of structural inequality on cancer prevention strategies. 

Limitation. please clearly address the limitation of the current review

1) as this is not the systematic review, may missing some informaiton from the systematic review process

2) Previous selected articles used in this study did not really measure the association between structural inequality and cancer-related AL, inflammation, stress, and lack of cancer prevention strategies; furthermore, it is unknown these structural inequality is always stronger in Black than Whites. 

there may be structural inequality in other races as well with the same issues of high stress, high inflammation, and risk of cancer. So please address that future studies also need to consider to examine structural inequality and cancer related health outcomes in other races as well. 

Author Response

REVIEWER 1

No further changes are requested.

RESPONSE: We thank the reviewer.

REVIEWER 2

Great work. However authors still need to better provide their efforts and quality of their work, instead of simply summarizing the multiple comments as very simple sentences. Also please do not highlight the revised part, please use Track change.  Also in the responses to comments, please provide page, and line of the revised part. so that we can easily match up the revised part corresponding to the reviewer comments. 

RESPONSE: We have used tracking and we provide page and line number.

I suggest the revision below:

  1. Barriers to Improving Health Outcomes – change title to “Impact of Structural Inequality on Cancer Prevention Strategies”

RESPONSE: We have changed the title to Section 5 to “Impact of Structural Inequality on Cancer Prevention Strategies” as suggested by the reviewer. (Page 9, line 292)

Limitation. please clearly address the limitation of the current review

1) as this is not the systematic review, may missing some information from the systematic review

2) Previous selected articles used in this study did not really measure the association between structural inequality and cancer-related AL, inflammation, stress, and lack of cancer prevention strategies; furthermore, it is unknown these structural inequality is always stronger in Black than Whites. There may be structural inequality in other races as well with the same issues of high stress, high inflammation, and risk of cancer. So please address that future studies also need to consider to examine structural inequality and cancer related health outcomes in other races as well. 

RESPONSE:

We have added a paragraph at the end of the last section addressing the limitations of our review and future studies to address the issues as suggested by the reviewer. (Page 10, lines 345-352)

We thank the reviewers for allowing us to improve our manuscript.

This manuscript is a resubmission of an earlier submission. The following is a list of the peer review reports and author responses from that submission.

Round 1

Reviewer 1 Report

Comments and Suggestions for Authors

Dear authors,

Thank you for submitting a manuscript on such an important topic. The manuscript provides a focused overview of the topic and the relevant arguments. It is written with a prompt structure and in an argumentatively comprehensible manner. In the following, I give some minor suggestions for improving your manuscript:

- There are double spaces in several places, for example in lines 51, 66, 115.

- Line 15. After a first look at the Abstract, I thought you had done a geographic analysis based on zip code. The reason for this is that this is the only detailed information in the otherwise very general wording. I therefore recommend that you omit this information from the Abstract.

- Line 68. What is an AL unit?

- Line 114. I am neither American nor do I live in America. I am under the impression that black Americans are a fairly large group of residents (in terms of population) and are neither a minority nor a marginal subculture. I agree that this sentence should be included in the manuscript, but I think it would be beneficial to the scientific paper (and to readers) if you would explain how Black Americans can be considered a minority and marginal subculture (due to systemic and also official barriers they face).

- Lines 123-127. I ask the authors to revise this sentence (revise what was done in the studies mentioned). If you do a statistical analysis with correction for one variable (race in this sentence), then you cannot observe the association for the same variable (race).

- Lines 166-171. The manuscript focuses on the risk of developing cancer. These lines focus on interventions after diagnosis/treatment of cancer. I do not object to this topic being included in the manuscript as well, but I think an additional transition to this topic would be beneficial to the reader. Also, this is a different, broad topic, and if it is included, it should be expanded to include other important points, such as late vs. early interventions (prior to completion of treatment), interventions aimed at earlier return to work, ... It is also unclear whether the paragraph in lines 172-178 is about cancer prevention or about improving quality of life after a cancer diagnosis.

- Lines 180-185. Literature should be added after the naming of the individual studies.

- Line 240. Chapter 5 deals more or less exclusively with the barriers to promoting physical activity among black Americans. The authors should also describe the barriers to other strategies described in Chapter 4.

- Lines 276-279. The following items are missing: author's contributions, funding, conflicts of interest.

I wish you much success in your future research.

Author Response

We thank the reviewer for allowing us to improve our manuscript. We have made the suggested changes as requested by the reviewer.

COMMENT AND RESPONSE:  We have eliminated the double spaces in several places, for example in lines 51, 66, 115.

COMMENT: Line 15. After a first look at the Abstract, I thought you had done a geographic analysis based on zip code. The reason for this is that this is the only detailed information in the otherwise very general wording. I therefore recommend that you omit this information from the Abstract.

RESPONSE: We have eliminated the Zip Code wording.

COMMENT: Line 68. What is an AL unit?

RESPONSE: We have added the description of the AL unit.

COMMENT: Line 114. I am neither American nor do I live in America. I am under the impression that black Americans are a fairly large group of residents (in terms of population) and are neither a minority nor a marginal subculture. I agree that this sentence should be included in the manuscript, but I think it would be beneficial to the scientific paper (and to readers) if you would explain how Black Americans can be considered a minority and marginal subculture (due to systemic and also official barriers they face).

RESPONSE: We have added a paragraph describing Black Americans as a minority population Lines 119-125..

COMMENT- Lines 123-127. I ask the authors to revise this sentence (revise what was done in the studies mentioned). If you do a statistical analysis with correction for one variable (race in this sentence), then you cannot observe the association for the same variable (race).

RESPONSE: We have revised this section and added statistical information (lines 133-147)

COMMENT: - Lines 166-171. The manuscript focuses on the risk of developing cancer. These lines focus on interventions after diagnosis/treatment of cancer. I do not object to this topic being included in the manuscript as well, but I think an additional transition to this topic would be beneficial to the reader. Also, this is a different, broad topic, and if it is included, it should be expanded to include other important points, such as late vs. early interventions (prior to completion of treatment), interventions aimed at earlier return to work, ...

RESPONSE: We have added additional information on the topic.

COMMENT: It is also unclear whether the paragraph in lines 172-178 is about cancer prevention or about improving quality of life after a cancer diagnosis.

RESPONSE: We have added clarification for both.

COMMENT: - Lines 180-185. Literature should be added after the naming of the individual studies.

RESPONSE: Done

COMMENT: - Line 240. Chapter 5 deals more or less exclusively with the barriers to promoting physical activity among black Americans. The authors should also describe the barriers to other strategies described in Chapter 4.

RESPONSE: We have added description of barriers to other strategies.

COMMENT: - Lines 276-279. The following items are missing: author's contributions, funding, conflicts of interest.

RESPONSE: We have added the information.

COMMENT: I wish you much success in your future research.

RESPONSE: Thank you!

Reviewer 2 Report

Comments and Suggestions for Authors

This is important review, but formal structure as a manuscript format is lacking. 

Overall, authors just wrote the existing literature where are already known and published in many previous studies, not sure this type of writing is belong to what type of research, is this will be a scoping review? Even though scoping review, further structure is needed

1. some parts authors mentioned about area-related discrimination, in inflammation, and cancer, and stress, However, it was not clear authors really cited articles showing inflammation, cancer risk, and stress, and environmental disparities in Black population. Without providing any correlation or associations test from the existing literature, it is hard to say these are all linked in Black. 

Seems like authors was writing this paper mostly based on hypothesis, by assuming AL, stress, discrmination, environmental disparities are all linked in Black.

This is not the research  proposal, there need to be cautious in writing the potential hypothesis.

2. Methods please provide structural methods inclusion/exclusion criteria, databases used, even though scoping review

3. Introduction What are known and what are not known. why this study is needed. 

4. Conclusion: what are the primary findings of this study is missin

5. discussion: what are the information newly added to the existing literature.

6. provide some tables to summarize the findings for readers to easily follow. 

Author Response

COMMENT: This is important review, but formal structure as a manuscript format is lacking. Overall, authors just wrote the existing literature where are already known and published in many previous studies, not sure this type of writing is belong to what type of research, is this will be a scoping review? Even though scoping review, further structure is needed

RESPONSE: We thank the reviewer for allowing us to improve our manuscript. We have added some sub-sections to improve the structure.

COMMENT: some parts authors mentioned about area-related discrimination, in inflammation, and cancer, and stress, However, it was not clear authors really cited articles showing inflammation, cancer risk, and stress, and environmental disparities in Black population. Without providing any correlation or associations test from the existing literature, it is hard to say these are all linked in Black. 

RESPONSE: We have added further references to define the linkage to Black population.

COMMENT: Seems like authors was writing this paper mostly based on hypothesis, by assuming AL, stress, discrimination, environmental disparities are all linked in Black. This is not the research  proposal, there need to be cautious in writing the potential hypothesis.

RESPONSE: We agree there is some element of hypothesis but we have added references to link it together.

COMMENT: Methods please provide structural methods inclusion/exclusion criteria, databases used, even though scoping review

RESPONSE: We have reviewed only relevant PubMed literature.

COMMENT: Introduction What are known and what are not known. why this study is needed.

RESPONSE: We have added a paragraph to explain it better. 

COMMENT: Conclusion: what are the primary findings of this study is missing

RESPONSE: We have revised the conclusion.

COMMENT: discussion: what are the information newly added to the existing literature.

RESPONSE: We have described the linkage between stress, inflammation and cancer.

COMMENT: provide some tables to summarize the findings for readers to easily follow. 

RESPONSE: We think adding a table will only duplicate the information presented.

Round 2

Reviewer 2 Report

Comments and Suggestions for Authors

Authors still did not fully address the reviewer's comments

Methods: in the abstract as well as method, where authors mention the methodology part? no information regarding the type of this study, database was only pubmed, what are the inclusion/exclusion criteria etc. 

even though scoping review, we should have method section in general.

Overall direction: I am still very confused about this study. I don't think the current study is applicable for peer-reviewer journal. as this study seems better as research brief, or opinion sections. as the current study did not fully follow the rigor of the manuscript.

Topic: AL, inflammation, cancer risk and racial/ethnic disparities are well known. I do not still see the what part of new information of this study is achieved. It seems like just summary some part of existing literature as case report., 

In abstract: While addressing 30 health disparities require multi-level solutions, comprehensive systemic reforms to address racism, 31 ensure equitable healthcare access, and foster environments conducive to healthy lifestyles are nec- 32 essary to enhance health outcomes

This sentence should be revised or deleted. as the current study are not systematic review, this study did not address the limitation of comprehensive understanding about this topic.

In TABLE, included studies did not really show the link with cancer.

there are literature showing inflammation, stress, race/ethniciy differences between non-cancer groups and cancer groups. I think this type of articles should be included and addressed, instead of general literatures.

Conclusion: I do not see any revision given my previous comments. Furthermore, this section

There is a complex interplay between structural inequity, chronic psychological 319 stress, inflammation, and cancer. The impact of systemic racism, historic policies such as 320 "redlining," and the resultant living conditions of many minoritized Americans contribute 321 to psychological stress or allostatic load.

The current section above needs to be changed. as the current study did not really address structural racisim, policies, this seems too extreme sentences, not related to the findings ? or summary of the current work.